# Improving Drug Delivery on Candida Albicans Using Geraniol Nanoemulsion

**DOI:** 10.3390/pharmaceutics15102475

**Published:** 2023-10-17

**Authors:** Cristiano Silva Pontes, Gabriel Garcia de Carvalho, Andressa Rosa Perin Leite, Marlus Chorilli, Denise Madalena Palomari Spolidorio

**Affiliations:** 1Department of Physiology and Pathology, School of Dentistry at Araraquara, São Paulo State University (Unesp), Araraquara 14801-903, SP, Brazil; timranho@gmail.com (C.S.P.); denise.mp.spolidorio@unesp.br (D.M.P.S.); 2Department of Dental Materials and Prosthodontics, School of Dentistry, São Paulo State University (Unesp), Rua Humaitá, 1680, Araraquara 14801-903, SP, Brazil; andressa.perin@unesp.br; 3Department of Drugs and Medicines, International School of Pharmaceuticals Sciences, São Paulo State University, Araraquara 14801-903, SP, Brazil; marlus.chorilli@unesp.br

**Keywords:** geraniol, nanoemulsion, *C. albicans*, biofilm, citotoxity, antifungal

## Abstract

Geraniol (GE) is a monoterpene alcohol with excellent antifungal activity. However, its low solubility and high volatility impair its use. Nanoemulsions (NE) are excellent delivery systems for poorly soluble and volatile drugs, achieving controlled release of the active ingredient. The aim of this study was to improve the delivery of geraniol (GE) incorporated in NE against *Candida albicans* in order to evaluate the antibiofilm effect and cytotoxicity. Nanoemulsion containing 10% oil phase (cholesterol) (*w*/*w*), 10% surfactant (mixture of soy phosphatidylcholine and Brij 58; 1:2) (*w*/*w*), and 80% aqueous phase (phosphate buffer) (*w*/*w*) was synthesized. Incorporation of GE was carried out by sonication and the final compounds were characterized by hydrodynamic diameter, polydispersity index (PDI), and zeta potential (ZP), in addition to evaluation of physicochemical stability after 6 months and 1 year. The GE-NE effect was evaluated on *Candida albicans* biofilms and cytotoxic effect was evaluated on immortalized normal oral cell line NOK-Si. The diameter of GE-NE was 232.3 ± 2.7 nm and PDI 0.155 with exhibited homogeneity and stability in solution. GE-NE showed antibiofilm activity at a concentration of 75 μg/mL with reduction of >6.0 log_10_, and no cytotoxicity against NOK-Si cells at concentrations below 150 μg/mL was observed. GE-NE proved to be a promising candidate for prevention and treatment of fungal diseases.

## 1. Introduction

*Candida albicans* is an opportunistic pathogen, polymorphic, capable of undergoing morphological transition between the forms of yeast, pseudohyphae, and hyphae [1], residing in the gastrointestinal, genitourinary, conjunctival, and oral tracts. It is considered the main species causing candidiasis and nosocomial infections with high mortality rates due to invasive candidemia and drug resistance [2,3].

In the oral cavity, *Candida albicans* is the most commonly isolated yeast species and can trigger infections under favorable conditions such as poor oral hygiene and presence of dental prostheses and devices, associated with its capacity to form biofilms, generating virulence factors and the development of its pathogenesis [4]. Biofilms are communities of microorganisms that are structured on biotic or abiotic surfaces incorporated into a matrix of extracellular polysaccharides providing a structural framework that favors metabolic exchanges and protection to the cells of this biofilm, making it difficult for the action of antimicrobials [5]. 

Azole derivatives, polyenes, echinocandins, and nucleoside analogues are the main drugs used in the treatment of most fungal infections. However, most of these agents have limited efficacy due to the development of resistant strains, host toxicity, and undesirable side effects, which compromise their use in medical and dental practice [6]. Mechanisms of antifungal resistance developed by microorganisms are based on the reduction in effective concentration of drugs with increased drug efflux mediated by membrane transport proteins, which target changes such as overexpression of sterol biosynthesis genes, and changes in metabolism to divert the toxic effects exerted by some antifungal agents [7,8,9]. In biofilms, resistance to antifungal agents is exacerbated, multifactorial, and mechanically complex, limiting the therapeutic potential [10]. 

Therefore, natural products are being investigated as promising agents for surface biofilm removal, prevention, and treatment [11]. Geraniol (GE) is the common constituent of several essential oils, such as citronella, and is widely used as a chemical in cosmetic fragrances and home use [12]. In addition to its low toxicity, its effect as a repellent and insecticide for pest control showed good chemopreventive activity, as well as antimicrobial, antioxidant, and anti-inflammatory properties [13]. However, GE showed high volatility exacerbated with increasing temperature, short action time, and high hydrophobicity [12].

Nanocarriers allow for better drug distribution to its site of action, a factor that provides a series of benefits to the patient, such as increased therapeutic efficacy, reduced doses, fewer administrations, and decreased side effects [14,15]. Nanoemulsion (NE) can be formed by dispersing oil in water or water in oil, allowing the incorporation of different active substances [16,17,18]. Owing to their benefits (such as thermodynamic stability, low cost, easy preparation, simple composition, and capability to carry hydrophobic substances), these nanocarriers have attracted interest from the scientific community [19].

Therefore, an NE system appears to be a safe alternative for the incorporation of GE in order to increase its action time, allowing for its slower release and guaranteeing a spreading on the surfaces [20]. In addition, it provides better biodegradability, tolerability, and bioavailability of the drug in order to guide its action to the specific site of interest [21]. In view of the problem described above, there is great interest in developing new strategies to combat and control fungal infection. This includes the search for new sources of bioactive molecules with antimicrobial capacity, pharmacological efficiency, and low levels of side effects. Considering the application of drug delivery systems as a strategy to improve the antimicrobial profile of bioactive compounds and based on promising studies using crude GE extract, the objective of this study was to evaluate the action of geraniol associated with a new nanoemulsion system against *Candida albicans* biofilm and cytotoxic effect on immortalized normal oral cell line NOK-Si. Although there are studies in the literature evaluating the antimicrobial effect of GE, none of these studies have considered the incorporation of GE in an NE against *Candida albicans* biofilms. In addition, there are few published data on GE cytotoxic assay [22,23], and the present study addresses this literature deficiency. Incorporation of GE was carried out by sonication and final compounds were characterized by hydrodynamic diameter, polydispersity index (PDI), and zeta potential (ZP), in addition to evaluation of physicochemical stability after 6 months and 1 year.

## 2. Materials and Methods

### 2.1. Nanoemulsion (NE) Preparation

The nanoemulsion was prepared according to Formariz et al. [24] and Bonifácio et al. [25] with modifications. The oil phase (OP) was composed of cholesterol (CHO) (10%; *w*/*w*); polyoxyethylene 20-cetyl ether (Brij 58^®^, CAS 9004-95-9, Sigma-Aldrich Brasil Ltda, São Paulo, Brazil) + soy phosphatidylcholine (PS-Epikuron^®^ 200, CAS 8002-43-5, Cargil, Germany), which was used as a surfactant (S) (2:1; 10%, *w*/*w*). The aqueous phase was composed of phosphate buffer (pH 7.4) (80%; *w*/*w*). In sequence, the mixtures were sonicated (Q700; QSonica, Newtown, CT, USA) at 700 watts power and 13% amplitude, in batch mode for 10 min, with an interval of 30 s every 2 min, in an ice bath. Geraniol (Sigma-Aldrich, St. Louis, MO, USA) 2 mg/mL was incorporated and sonicated similarly to NE, but at an amplitude of 17%. The blank formulation was prepared as described but without added geraniol. All sonifications were performed twice, unlike the methodology described earlier [24,25].

### 2.2. Nanoemulsion Characterization

#### Average Hydrodynamic Diameter, Polydispersity Index, and Zeta Potential

The preliminary stability of the formulations was analyzed through organoleptic characteristics (aspect, opacity/translucency), droplet size, polydispersity index, and zeta potential. Twenty-four hours after the preparation of formulations, indications of instability were evaluated by observing signs of creaming or phase separation, along with physical aspects such as translucency, fluidity, bluish reflection, and odor. After encapsulation of geraniol in NE (GE-NE), the formulations were diluted in ultrapurified water (1:1000; *v*/*v*) in triplicate at 25 °C and characterized for zeta potential through eletrophoretic mobility, average hydrodynamic diameter, and polydispersity index (PDI) through dynamic light scattering (DLS) using Zetasizer equipment (Nano ZS, Malvern Instruments Ltd., Malvern, UK). To prevent Ostwald ripening, coalescence, and flocculation, the formulations were stored at 4 °C [26,27], and after six and twelve months, the NEs were again evaluated to check stability and possible changes in the diameter of particles. Zeta potential (ZP) is an indication of repulsive forces between oil droplets in an emulsion. High ZP values (±30 mV) indicate the difficulty of droplet coalescence and a high stability of emulsion.

### 2.3. Transmission Electron Microscopy (TEM)

The geraniol-loaded nanoemulsion was diluted (1:20) in deionized water and 3 µL was added to the copper grids, excess was gently removed with absorbent paper and counterstained for 5 min with 1 drop (~4 µL) of an acetate solution (2% uranyl acetate solution (*v*/*v*) in distilled water). The excess sample was removed again with filter paper, and dried at room temperature for 15 min before analysis. Transmission electron microscopy of the samples were performed using a Phillips CM200 Transmission Electron Microscope, operating at 200 kV and with a LaB6 (lanthanum hexaboride) filament [28].

### 2.4. Microbial Strain and Antimicrobial Solutions

#### 2.4.1. Solutions

Geraniol (Sigma-Aldrich, St. Louis, MO, USA; CAS: 106-24-1) was prepared at concentrations ranging from 1.2 to 0.015 mg/mL [29] in Roswell Park Memorial Institute (RPMI 1640, Acumedia, Lansing, MI, USA) medium and 0.4% dimethyl sulfoxide (DMSO) (Sigma-Aldrich, St. Louis, MO, USA; CAS: 67-68-5). A stock solution of Nystatin (Sigma-Aldrich, St. Louis, MO, USA; CAS: 1400-61-9) at a concentration of 5 mg/mL was prepared in DMSO and subsequently diluted from 0.64 to 0.012 mg/mL in RPMI-1640 medium [29].

#### 2.4.2. Reactivation and Preparation of *C. albicans* Suspension

*Candida albicans* (American Type Culture Collection, ATCC 900028; Rockville, MD, USA) was grown in 5 mL of RPMI 1640 culture medium supplemented with 2% glucose (CAS: 50-99-7) and 0.165 M MOPS (3-(N-morpholino); CAS: 1132-61-2) propanesulfonic acid at pH 7.2 and then incubated at 37 °C for 24 h. The suspensions were adjusted to a final concentration of 10^7^ CFU mL^−1^ using a spectrophotometer (OD = 600 nm, Eppendorf AG, Hamburg, Germany).

#### 2.4.3. Minimum Inhibitory Concentration (MIC) and Minimum Fungicidal Concentration (MFC)

The MIC was determined using the microdilution technique in 96-well plates according to the methodologies described in accordance with the Clinical and Laboratory Standards Institute (CLSI) [30]. The microplate wells were filled with 100 μL of RPMI- 1640 buffered with MOPS, and then followed by 100 μL of serial dilutions of GE and GE-NE in concentrations of 1200 to 2.35 μg/mL in the microplate wells. The fungal suspension (5 μL) of 2.5 × 10^3^ CFU mL^−1^ prepared in RPMI was added to each well and the plates were incubated at 37 °C for 24 h. *Candida albicans* without treatment was used as a control group, and Nystatin was used as a positive control at concentrations 64 to 0.0625 μg/mL (ten times the serial dilution) according to the CLSI protocol [30]. A DMSO group was used to demonstrate that this solution did not interfere with the results. After 24 h, the wells were evaluated visually and by using a microplate reader at 600 nm (Synergy H1, Biotek). The MIC was defined as the lowest concentration that would completely inhibit growth, based on spectrophotometry analysis.

Subsequently, the minimum fungicidal concentrations were established. Briefly, 10 µL samples were transferred to a Petri plate with Sabouraud Dextrose Agar culture medium (Acumedia, Lansing, MI, USA) and the cultures were maintained at 37 °C for 48 h. The MFC was defined as the lowest concentration that would reduce *Candida* species by ≥99.9%. All experiments were performed in triplicate and in three independent experiments.

### 2.5. Antibiofilm Evaluation

#### 2.5.1. Human Saliva Preparation

The present study was previously reviewed and approved by the Ethics Committee of the School of Dentistry at Araraquara, São Paulo State University (Unesp), Brazil, CAAE: 46795821.6.0000.5416. Saliva was processed according to a previous study [31].

#### 2.5.2. Biofilm Formation and Biofilm Analyses by Colony-Forming Units (CFU)

Biofilm formation was carried out according to previous studies [32,33]. Equal volumes of cell suspensions of *C. albicans* (1.0 × 10^6^ CFU mL^−1^) were transferred to wells of the 96-well plate and kept at 37 °C in an orbital shaker at 75 rpm for 1 h and 30 min at 37 °C, which corresponded to the adhesion phase. Plates were pretreated with 50 μL of saliva for 1 h. Then, the suspension was aspirated and the wells were washed with sterile PBS (phosphate-buffered saline) to remove nonadhered microorganisms. Then, 100 μL of RPMI 1640 medium was added to each well and kept at 37 °C for 48 h at 75 rpm to promote the biofilm growth.

After biofilm formation, the culture medium was removed and 100 μL of the concentrations of GE and GE-NE test solutions was added separately to *C. albicans* biofilms. Nystatin (640 μg/mL) was used as a positive control and untreated wells were used as a negative control. After 24 h, resulting biofilms in each well were washed twice and the mass of the biofilm was carefully scraped in the presence of 100 μL of PBS. The resulting suspension was diluted and inoculated in medium Sabourad dextrose agar and plates were kept at 37 °C for 24 h; the CFU/mL^−1^ was then calculated.

#### 2.5.3. Confocal Scanning Laser Microscopy (CLSM)

The effect of the NE-GE on biofilm was visualized in CLSM imaging. Samples with treated biofilms were stained with the LIVE/DEAD BacLight Bacterial Viability Kit (L7012, Invitrogen Molecular Probes, Eugene, OR, USA), according to the manufacturer’s instructions. Excitation/emission wavelengths were 488/500 nm for SYTO 9 and 488/635 nm for Propidium Iodide. The fluorescence of the stained cells was visualized by CLSM (LSM 780, Zeiss, Jena, Germany), and the images were acquired using ZEN 2012 software (Zeiss) at 1024 by 1024 pixel resolution. The images of a single focal plane of the biofilm were captured by the system with a 20× magnification lens. For image acquisition, the groups whose previous tests provided greater microbial elimination were used.

#### 2.5.4. Morphological Evaluation of the Biofilm by Scanning Electron Microscopy (SEM)

*Candida albicans* were seeded under round glass coverslips (13 mm in diameter) introduced into 24-well flat-bottomed polystyrene plates. The incubation procedures were the same as for the CFU counting experiment, until the time of washing with PBS. After biofilm treatment and using the same groups tested in CFU analysis, the coverslips were immersed in 2.5% glutaraldehyde for 1 h for the attachment of microorganisms, followed by three washes with PBS, and gradual dehydration in a series of alcohols (70, 90% for 1 h each and five washes of 15 min with absolute alcohol). Coverslips were dried at room temperature and placed in a vacuum desiccator with silica gel for 7 days.

Subsequently, the samples were metallized with gold–palladium and fixed to the stubs using a conductive adhesive and the morphological analysis of the samples was performed using a Zeiss electron microscope (Jeol, model JSM-7500F) with a voltage of 5.00 kV.

### 2.6. Cytotoxicity Assay

Spontaneously immortalized normal oral keratinocytes (NOK-SI-RRID: CVCL-BW57) from human oral mucosa were seeded (1 × 10^5^ cells/well) on culture medium Dulbecco Mem/Ham F-12 medium (Sigma-Aldrich, MO, USA) supplemented with 10% fetal bovine serum (FBS) (Cultilab, São Paulo, Brazil) and 1% penicillin, streptomycin, and glutamine (100 IU/mL of penicillin, 100 μg/mL of streptomycin, and 2 mmol/L glutamine—Gibco, NY, USA). The plates containing cells in the wells were placed in an incubator at 37 °C with 5% CO_2_ for 24 h. The extracts were added and again incubated for 24 h according to ISO 10993-522 [34]. After this period, viability assays (Alamar Blue and Live/Dead) were performed.

The extracts tested were prepared by diluting the GE in DMEM culture medium with supplementation by 1% FBS at final concentrations of 150 to 600 μg/mL. NEs were diluted in DMEM medium with 1% FBS supplementation in a proportion of 1:10 (*v*/*v*); all groups were incubated at 37 °C for 24 h according to ISO 10993-12:2009. After this period, supernatants were filtered with 0.22 µm^2^ membranes, and the solutions obtained were diluted in DMEM supplemented with 1% FBS to reach the following concentrations for blank-NE (100%, 50%, 25%, 12.5%, and 6.25%) and GE-NE (1200, 600, 300, and 150 μg/mL).

The cell viability was analyzed by Alamar Blue assay (n = 8) that detect cellular metabolic activity by means of a fluorimetric indicator (AlamarBlue^TM^ HS Cell Viability Reagent; Invitrogen, Waltham, MA, USA). The corresponding 10% of the Alamar Blue solution culture volume was added to the culture medium with 1% FBS for 4 h in contact with the cells in an incubator. Fluorescence was verified by using a fluorimeter at 570/600 nm (Synergye H1; BioTek Winooski, Winooski, VT, USA) [30].

The live/dead assay was carried out to evaluate qualitative cell viability (n = 2). This analysis was performed by incubating the cells with 4 μM of ethyl homodimer-1 (dead Eth-1 cells = red fluorescence) and 2 μM of Calcein AM (viable cells CA = green fluorescence) in nonserum DMEM at room temperature for 45 min (Kit Live/Dead^®^ Viability/Cytotoxic; Invitrogen, Waltham, MA, USA) and were evaluated using a Leica DM 5500B fluorescence microscope (Nussloch GmbH, Nussloch, Germany) [31].

### 2.7. Statistical Analysis

After sample distribution and homoscedastic analysis, data were evaluated by one-way ANOVA with Tukey’s post hoc test. All analyses were performed at α = 0.05 using SPSS software (v.17, SPSS Inc, Chicago, IL, USA). For the characterization of NE, CLSM, SEM, and fluorescence cell viability, descriptive analyses were performed.

## 3. Results

### 3.1. Characterization of Nanoemulsion

Twenty-four hours after the preparation of formulations (GE-NE and NE), changes were not observed in organoleptic characteristics and signs of instability. The formulations presented an opaque and milk characteristic (Figure 1) and showed the same characteristics as in the preliminary stability tests: fluid emulsion and absence of creaming or phase separation. There is a relationship between the size of the particles and their color, the smaller the size of the droplets, the clearer or more transparent they appear. White or milky colors indicate larger particles [35,36].

DLS was used to determine the average hydrodynamic diameter and polydispersity index for the formulated NEs. Potential zeta was verified by eletrophoretic mobility. Table 1 shows mean values and standard deviation of hydrodynamic diameter and PDI of formulations tested in this study, where it is observed that the diameter of NE particles (blank) was 150.8 ± 1.7 nm, and 232.3 ± 2.7 for GE-NE. PDI is an index that assesses the relative homogeneity in the sizes of particles distributed in samples, and it was calculated by dividing average droplet size by average number of measured droplets. For GE-NE, light-scattering analysis showed mean PDI values of 0.155 after formulation and monomodal distribution indicating a good droplet size distribution and stability of the NE system. This parameter directly reflects droplet size homogeneity in the NE.

#### 3.1.1. Stability of GE-NE over Six and Twelve Months after Formulation

Stability of emulsion-based delivery systems during storage and use is an essential requirement for their clinical use. Figure 2 shows the results for the evaluation of the particle diameter and PDI from GE-NE at 6 months and 1 year following its formulation; small size variation, 1 and 5.7 nm, respectively, and a dispersion increase of 0.03 indicate that the NEs remained stable and homogeneous through time.

Zeta potential of GE-NE at the time of formulation (ZP = −3.3 mV, Figure 2) after 6 months (ZP = −4.60 mV, Figure 2) and 12 months (ZP = −7.38 mV, Figure 2) showed a small increase, indicating stability in the surface charge and homogeneity of system after prolonged storage of 1 year.

#### 3.1.2. Transmission Electron Microscopy (TEM)

In Figure 3, it is possible to observe that the droplets of blank nanoemulsion and droplets of geraniol-loaded nanoemulsion present similar characteristics, with a circular shape and sizes smaller than 200 nm.

### 3.2. MIC and MFC of GE and GE-NE under Planktonic Cultures of Candida albicans

Table 2 shows the antimicrobial effect of GE and GE-NE on the planktonic cultures of *C. albicans*. The incorporation of geraniol in nanoemulsion reduced MIC and MFC by 16 times against strains of *C. albicans*, compared to the pure extract of GE. Since no fungal growth was observed, the MFC/MIC ratio (2:1) demonstrated the fungicidal nature in all groups.

### 3.3. Quantification of the Biofilm

Figure 4 illustrates the effect of GE, GE-NE, and Nystatin on biofilm growth by quantifying viable cells (CFU mL^−1^). GE-NE (75 µg/mL) showed a significant reduction (*p* = 0.001) in biofilm (7.2 Log_10_ reduction) compared to the negative control, which was similar to the 1200 µg/mL concentration of pure GE and Nystatin extract (640 µg/mL). GE-NE treatment potentiated 16 times the antifungal effect of the GE extract. Nystatin showed antimicrobial activity, but the amount needed to eliminate the biofilm was 80 times greater (80× MIC) than its minimum inhibitory concentration.

### 3.4. Confocal Laser Scanning Microscopy (CLSM)

In Figure 5, the GE 300 µg/mL and GE-NE 75 µg/mL concentrations were chosen to illustrate the reduction in *C. albicans* biofilms using CLSM. Analysis of live/dead images of *C. albicans* biofilms demonstrated that all treatments affected the viability of biofilms, but greater thickness biofilm reduction was observed in the groups where GE-NE 75 µg/mL was applied. Biofilm thickness measurements are also indicated in Figure 5.

### 3.5. Scanning Electron Microscopy (SEM)

Scanning electron microscopy images (Figure 6A) showed the classic morphology of yeasts from the control group, showing a cluster of well-preserved cells, with the presence of hyphae and characterized by intact cell walls with a clear and smooth fibrillar outer layer [37]. Cells treated with GE 300 μg/mL showed altered and shrunken surfaces and discrete biofilm disintegration (Figure 6C). *Candida albicans* cells treated with GE-NE 75 μg/mL showed extensive and significant biofilm disintegration (Figure 6D), shrunken cells and altered cell-wall morphology, lacking the fibrillar layer, and disruption in the transition from yeast to hyphae, similar to Nystatin 640 μg/mL (Figure 6B).

### 3.6. Cytotoxicity Assay

Cytotoxic potential for GE and GE-NE on NOK-SI cells was evaluated by cytotoxicity assay and concentrations; cell viability results ≥ 70% were considered viable (Figure 7). The treatment with GE ≤ 150 μg/mL showed 90% viable cells. However, there was a cytotoxic effect at the GE concentration >300 μg/mL, whose number of viable cells was 52.61%, followed by smaller percentages for the other concentrations, with complete death occurring at GE > 1200 μg/mL, with a statistically significant decrease when compared with the negative control group (*p* < 0.001). Thus, GE concentrations ≤ 300 μg/mL were selected for antibiofilm studies. The same result of cytotoxic effect can be observed in GE-NE, there was a decrease in the percentage of viable cells at concentrations greater than 150 µg/mL, with complete death occurring at concentrations above 600 µg/mL.

## 4. Discussion

In this study, nanoemulsions were developed and the main physicochemical parameters were examined. The nanoformulations were kinetically stable, as they did not show gravitational separation [38]. The sonication method was used because it is effective, allowing better control of droplet size in nanoscale (30–600 nm), which is dependent on operation conditions (time and temperature) and a wide choice of formulation constituents [21,39,40].

Lecithin is a nontoxic amphoteric surfactant that is well tolerated by the body and it can be fully metabolized. It was classified by the Food and Drug Administration (FDA) as a safe product for human consumption and has been used as a surfactant for the production of NEs, since this compound consists of a mixture of phospholipids of animal or plant origin, biocompatible and biodegradable, with no charge at physiological pH [40]. In addition, lecithin (PS) contains negatively charged polar lipids that are of great importance in the long-term stability of emulsions, as they confer a negative zeta potential at the interface, allowing electrostatic repulsion between the dispersed droplets.

Brij 58 was used in this study due to its nonionic nature, high hydrophilic–lipophilic balance (HLB) value, and low critical micelle concentrations (CMC). In addition, nanoemulsion systems formed with mixed surfactants offer superior properties to pure components [41,42], and surfactants (especially nonionic) reduce interfacial system damage [43] and are less irritating than ionic compounds [44].

In this study, the average particle sizes corroborate other studies that consider that the size of the NEs must be between 10 and 250 nm and that a narrow size distribution is generally considered stable against the processes of flocculation, creaming, sedimentation, and coalescence [21]. When comparing the unloaded (blank) and loaded (GE-NE) formulations, an increase in particle size was observed after incorporation, which is an indication of carryover in the lipid system.

PDI is an index that assesses the relative homogeneity of particle sizes distributed in the sample [45], and narrow values are necessary for its high stability. For the loaded and unloaded formulations, the light-scattering analysis showed average PDI values in the range 0.1 to 0.5 after formulation, indicating the homogeneity of droplet size and system stability [46], and corroborating other findings that obtained the same correlation [47,48].

Zeta potential (ZP) is a useful tool for estimating the stability of emulsions because it determines the electrostatic repulsion between globules. The ZP values obtained from the unloaded and loaded formulations showed a negative charge, and there was a small decrease in ZP after the incorporation of the GE, which may have been caused by the phenomenon of adsorption of the GE at the nanoemulsion interface, and by the presence of soy phosphatidylcholine (lecithin), which can provide negative charges to the particles [49,50].

Stability of GE-NE concerning particle diameter and PDI after 6 to 12 months was observed in this study. Stability of nanoemulsions is an essential requirement for their application as a drug delivery system. The stability study provides indications on the behavior of the product subjected to environmental conditions, from manufacture to expiration date. Stability tests are considered a predictive procedure for conditions that accelerate changes that may occur in market conditions. Owing to conditions in which it is conducted, this type of test is not intended to estimate the shelf life of the product, but rather to assist in the screening of formulations throughout production before reaching the final product [51].

Minimum inhibitory concentration assay provides information on the best useful concentration to exert an antimicrobial activity against the tested microbial strain, and lower values correspond to greater potential in production of an active compound. The product shows strong antimicrobial activity when it has a MIC between 50 and 500 μg/mL, moderate activity with a MIC between 600 and 1500 μg/mL, and weak activity with values over 1500 μg/mL [52,53,54].

In this work, results of MIC and MFC of geraniol in planktonic cultures of *C. albicans* demonstrate that GE is effective against the *fungi* in suspension. The concentrations found are similar to those described in the literature [55] and more effective than one reported elsewhere [56]. However, our results differ from the results obtained in other studies where geraniol inhibited *C. albicans* and non-*albicans* strains with MIC between 16 and 130 μg/mL [28,57]. Variability in the values of MIC and MFC in the literature may be due to variations in the strains and solvents used. The reduced MIC of GE-NE can be explained by the presence of cholesterol in the composition of the NE system [25,58], allowing interaction with the ergosterol present in the fungal cell membrane, inducing the release of GE directly on the target. In addition, surfactants can cause the breakdown of lipids in the biological membrane, leading to a reduction in the structure of the membrane barrier and causing an increase in the permeation of active compounds [59,60].

The nature of the antimicrobial effect produced by a substance against a specific microorganism can be classified according to the ratio between its MFC and MIC, where a MFC/MIC ratio ≤ 2:1 indicates a fungicidal characteristic of the product, and a MFC/MIC ratio > 2:1 suggests this product is fungistatic [61]. Our result confirms another study reporting that GE had a fungicidal action against strains of *C. albicans* [28]. Results for the control group showed no inhibition of fungal growth. Research on fungicidal substances is important for implementation in clinical practice as it predisposes less risk to the acquisition of resistance by microorganisms and allows a better therapeutic effect in immunosuppressed individuals [62,63].

This study evaluated the action of GE and GE-NE on biofilms of *C. albicans*, causing complete reduction in mature biofilms. When compared with Nystatin used as a positive control in our study, a concentration of 80× MIC was necessary to eliminate the biofilm. Studies show that the effective antibiofilm concentration can be 10–1000 times higher than that of planktonic cultures (MIC) [32,64,65]. Therefore, our findings suggest a potential use for GE-NE as an antifungal in patients affected with chronic candidiasis.

Studying the mechanism of action of antifungals is a relevant strategy to limit the emergence of resistance to currently available drugs, as well as for the development of more powerful and safer drugs against infection [66,67]. Recent research has shown that treatment with GE weakens the activities of *Candida* spp., and that oral use of GE promotes inhibition of fungal biofilm formation and hyphae formation. The action of GE destroyed cell-wall function, and reduced plasma membrane activity (ATPase) and ergosterol levels, suppressing fungal expansion. In addition, GE affected mitochondrial function and iron homeostasis and mitigated genetic toxicity [68,69].

Another study concluded that GE altered the sterol of *Candida* species at sub-MIC concentrations, and completely blocked ergosterol biosynthesis at MIC values, similar to fluconazole (FLC). FLC and other azoles inhibit the enzyme cytochrome P450—14a-demethylase, which prevents the formation of ergosterol from lanosterol. Therefore, inhibition of *Candida* cell growth by GE may be due to reduced levels of ergosterol, which is an essential component in the fungal cytoplasmic membrane, interrupting the integrity of this membrane and its homeostasis. The plasma membrane contains a protein that pumps H+ out of the cell, generating a proton motive force that helps in the intake of nutrients. Glucose-induced acidification of the extracellular medium by fungal cells is a measure of PM-ATPase-mediated H+ pumping [57], the effect of GE on PM-ATPase and ATPase-mediated H+ pumping, which is essential for fungal survival, was observed in some studies. In addition, drugs such as FLC and amphotericin B have no significant effects on H+ ATPase [57,70].

A literature review revealed that the nonpolar character of GE makes it possible to modify the lipid structure of the microorganism’s cell membrane, altering homeostasis and making it more susceptible to penetration by antibiotics [71]. In our investigation, the nanoscale and controllable size of nanoemulsions associated with GE incorporation resulted in new properties such as bioavailability and greater interaction with cells due to increased surface area, causing greater contact of the GE-NE globules around the hyphae and fungal spores and consequent potentiation of the antimicrobial and antibiofilm effect of the terpenoid [72].

In order to prove the possible effects of GE and GE-NE at the cellular level of *C. albicans* biofilms, scanning electron microscopy (SEM) analyses were performed. The main cellular modifications induced by treatments were alterations in cell wall and death. Our results were similar to another study that evaluated the morphology of untreated cells or cells exposed to GE, mainly affecting the cell wall and membranes of yeast [57].

The cytotoxicity of GE and GE-NE in host tissues was evaluated using an in vitro cell culture model of the human keratinocyte cell line NOK-SI. Cytotoxic assessments provide adequate understanding of the effective and nontoxic dose; it helps in determining the dose regimen. The maximum concentrations tested in this study were 600 μg/mL (2× MIC), and the results showed that GE concentrations lower than 150 μg/mL (MIC/2) indicated more than 90% viable cells; at a concentration of 300 μg/mL (MIC), the percentage of viable cells was 52.61%. Similar results were obtained in a study that evaluated cell viability in human peripheral blood mononuclear cells (PBMC), where concentrations less than or equal to 100 µg/mL showed more than 90% viable cells, and there was a drastic reduction in cell viability from the concentration of 250 µg/mL, and complete death was observed at 1000 µg/mL [23].

However, when we evaluated the effect of different concentrations of GE-NE on NOK-SI cells, there was a decrease in the percentage of viable cells at concentrations greater than 150 μg/mL followed by lower percentages for the other concentrations; complete death occurred at concentrations above 1200 μg/mL. The higher cytotoxicity of GE-NE on NOK-SI cells compared to the GE solution was possibly caused by the reduced size of the nanoparticles, resulting in an increase in surface area and greater cellular absorption of GE associated with the action of surfactants that can increase cell membrane permeability and assist in the absorption of the active ingredient, increasing its bioavailability [73]. It is important to note that NE (blank) did not exhibit an antimicrobial and antibiofilm effect, and showed no cytotoxic effect when diluted in the medium (50–12.5%) on NOK-SI cells, suggesting that the activity was manifested only by the GE and ruling out the possibility of disturbance of the other components in the nanostructured system.

Therefore, data obtained in our study allow us to infer that the nanostructured lipid system acted as a highly effective carrier of GE, providing better antifungal and antibiofilm activity on *C. albicans* strains. The NE we developed can be safely applied to other systems and provides better biological response. Our results confirmed the hypothesis that the incorporation of natural compounds into nanostructured lipid systems for drug delivery potentiates their action compared to their free form [74,75].

Nanotechnology applied in healthcare is constantly expanding and arousing increasing interest; based on results of this study, further research on the effect of GE and GE-NE for a wide range of pathogenic fungi and bacteria is necessary, in addition to in vivo toxicological studies, which would thus confirm the efficacy and safety of the compound for further clinical application. Future studies with the geraniol-release profile will be carried out and may complement this study.

## 5. Conclusions

The nanoemulsion showed promising results for preliminary stability tests and for physicochemical characterizations. Carrying GE into NE improved its antimicrobial and antibiofilm activity in vitro on *C. albicans* strains (75 μg/mL), suggesting the possibility for this system to become a potential therapeutic alternative to treat *Candida* infections. In cytotoxicity analysis, no significant effect on NOK-SI cells was observed for geraniol and nanoemulsion.

In summary:

Nanoemulsions showed average particle sizes of 150.8 ± 70.4 nm (NE blank) and 232.3 ± 122.5 nm (GE-NE).

Long-term stability of the GE-NE was evaluated, and no significant changes were observed in relation to particle diameter and PDI after 6 to 12 months. The range corresponding to the zeta potential for GE-NE at time of formulation (ZP = −3.3 mV), after 6 months (ZP = −4.60 mV), and after 12 months (ZP = −7.38 mV) showed only small variations, indicating the stability of system during the analyzed period of 12 months.

The incorporation in the nanostructured system improved the antifungal activity of the GE, presenting MIC of 18.75 μg/mL and MFC of 37.5 μg/mL, presenting a potentiated biological effect about sixteen times in relation to the simple extract of the GE. MFC/MIC ratio (2:1) of geraniol and GE-NE showed strong antifungal activity for the strain tested.

Antifungal activity was observed at concentrations of 1200 μg/mL of GE (4× MIC) and 75 μg/mL of NE-GE (4× MIC), causing complete reduction in mature biofilms.

From the analysis of images acquired through confocal laser microscopy, it was possible to verify that GE and GE-NE influenced the viability of simple *C. albicans* biofilms, corroborating the results of microbiological quantification analyses. Red marking showed cell death inhibiting the formation of simple mature *C. albicans* biofilms after treatment.

From the analysis of SEM, the images of *Candida* cells showed morphological changes in the presence of GE and GE-NE, however, with greater disruption and deformation of the cell wall resulting from the loss of cytoplasmic content of *C. albicans* cells treated with GE-NE compared to untreated cells that remained intact.

In cytotoxicity analysis, no significant effect on NOK-SI cells was observed for geraniol and nanoemulsion.

## Figures and Tables

**Figure 1 pharmaceutics-15-02475-f001:**
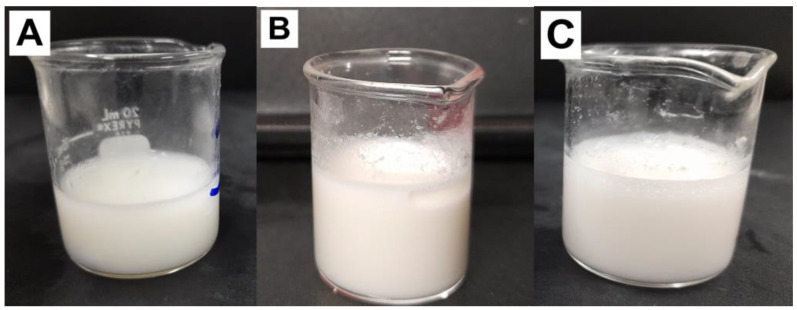
Representative images of nanoemulsion loaded with geraniol after (**A**) 0 h, (**B**) 24 h, and (**C**) 1 month.

**Figure 2 pharmaceutics-15-02475-f002:**
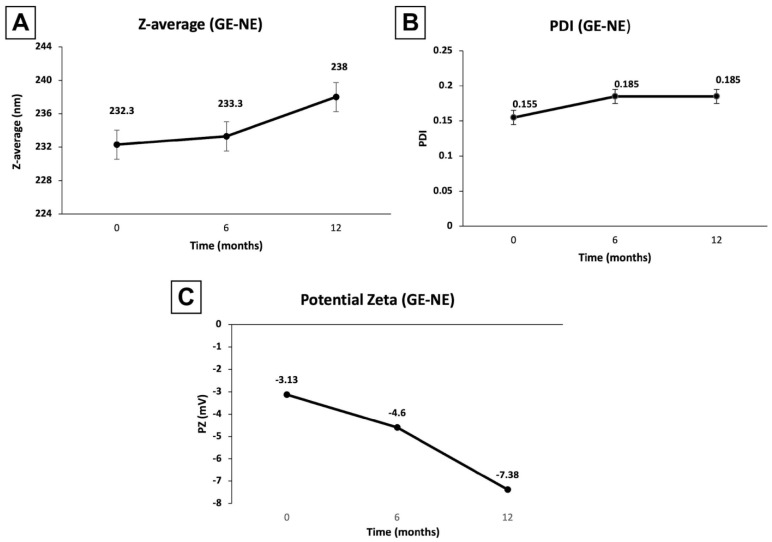
Droplet size (**A**), polydispersity index (**B**), and zeta potential (**C**) of geraniol-loaded nanoemulsion at the time of formulation, and after 6 and 12 months following synthesis.

**Figure 3 pharmaceutics-15-02475-f003:**
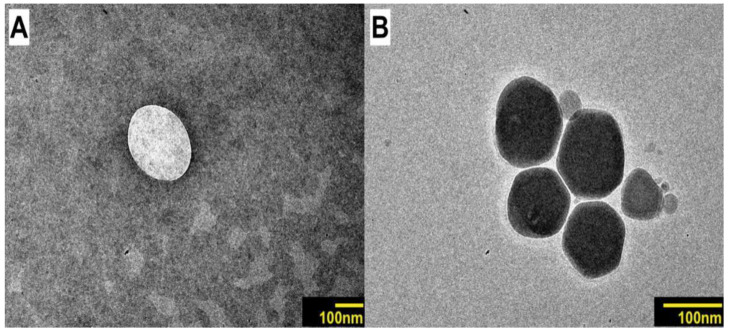
Images obtained by transmission electron microscopy (TEM): (**A**) blank nanoemulsion; (**B**) geraniol-loaded nanoemulsion.

**Figure 4 pharmaceutics-15-02475-f004:**
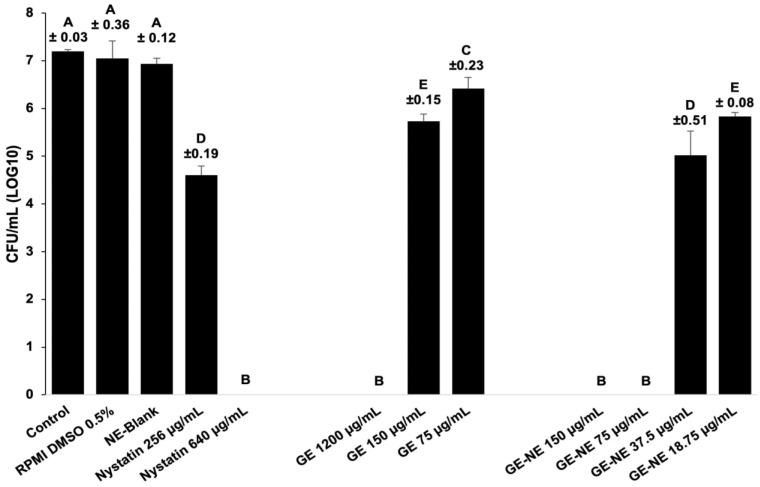
Quantification (CFU mL^−1^) of *C. albicans* biofilm after treatment: GE, GE-NE, and Nystatin. Different letters indicate statistical differences between groups (one-way ANOVA test, Tukey post test, *p* < 0.05).

**Figure 5 pharmaceutics-15-02475-f005:**
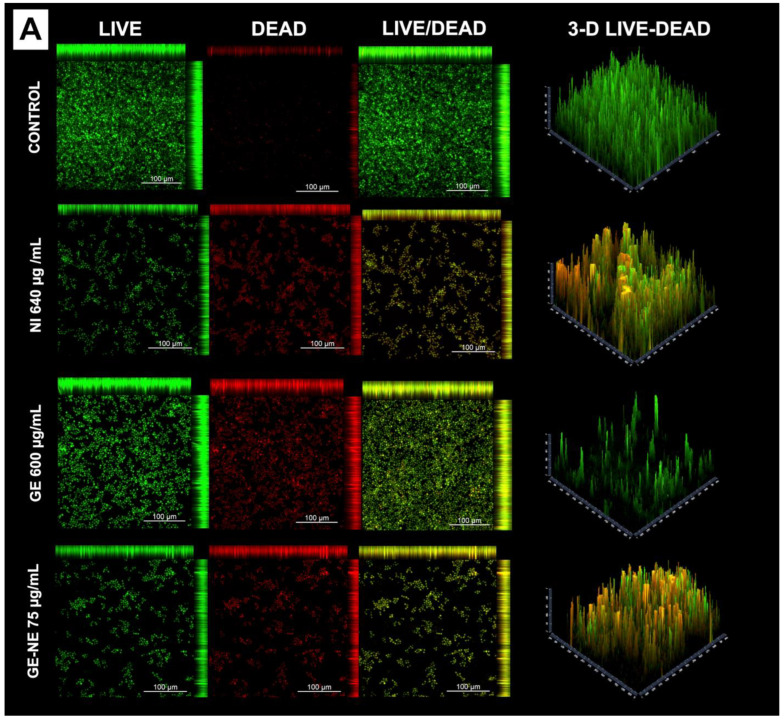
(**A**) Representative CLSM images of *C. albicans* biofilm: Nystatin 640 μg/mL, geraniol 600 μg/mL, and geraniol-loaded nanoemulsion 75 μg/mL. Dead cells are stained red and live cells are stained green. (**B**) Results are expressed as the means ± SD of triplicate assays for three independent experiments (ANOVA/Tukey test, α = 0.05). Different letters mean statistical difference.

**Figure 6 pharmaceutics-15-02475-f006:**
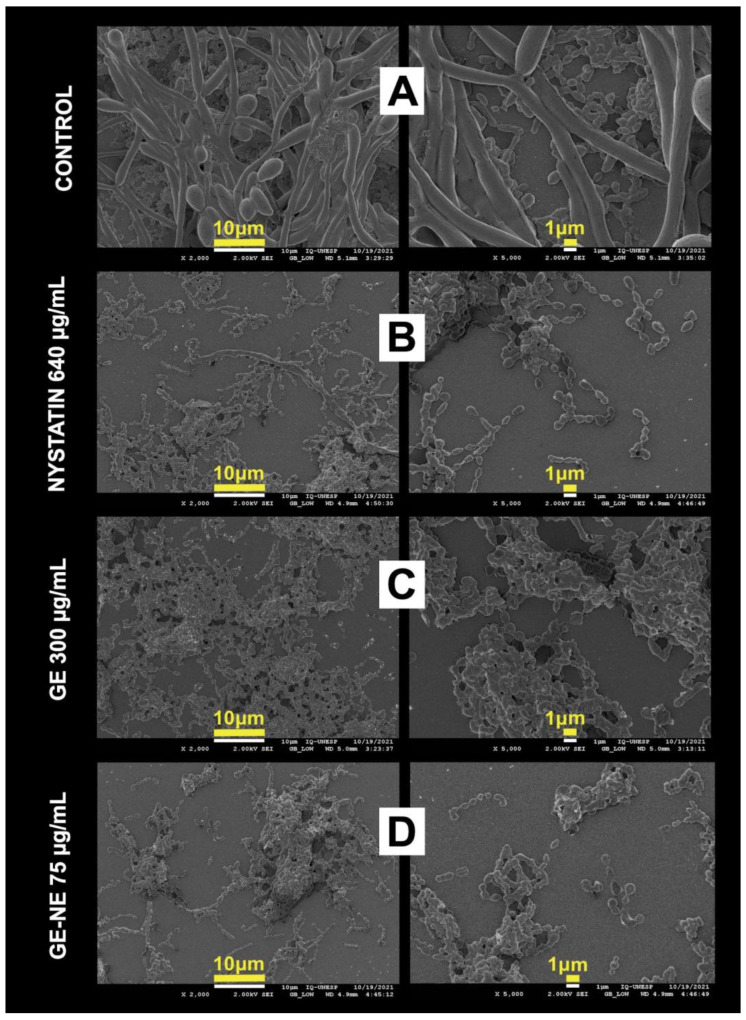
SEM images of simple biofilms of *C. albicans* after treatment with geraniol and nanoemulsions (2000× and 5000× magnifications; scale bars represent 10 µm and 1 µm): (**A**) control; (**B**) biofilm treated with Nystatin 640 μg/mL; (**C**) biofilm treated with GE 300 μg/mL; (**D**) biofilm treated with GE-NE 75 μg/mL.

**Figure 7 pharmaceutics-15-02475-f007:**
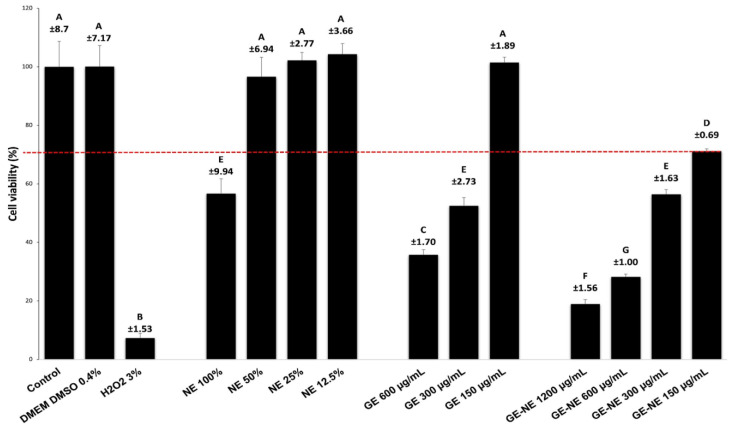
Cell viability percentage for oral keratinocytes NOK-SI cells after treatment with different concentrations of geraniol. The dotted line represents 70% viability. Equal letters represent statistical similarity between groups (one-way ANOVA test, Tukey post test, *p* < 0.05).

**Table 1 pharmaceutics-15-02475-t001:** Particle size, polydispersity index, and zeta potential value for NE and GE-NE.

Formulation	Particle Size (nm)	Polydispersity Index (PDI)	Zeta Potential (mV)
NE	150.8 ± 70.4	0.450	−6.35
GE-NE	232.3 ± 122.5	0.155	−4.60

**Table 2 pharmaceutics-15-02475-t002:** Minimum inhibitory concentration (MIC) and minimum fungicidal concentration (MFC) for GE, GE-NE, and Nystatin on the planktonic culture of *C. albicans*.

*C. albicans*			
	MIC	MFC	Nature
GE	300 μg/mL	600 μg/mL	Fungicide *
GE-NE	18.75 μg/mL	37.5 μg/mL	Fungicide *
Nistatina	4 μg/mL	8 μg/mL	Fungicide *

* MFC/MIC ratio 2:1.

## Data Availability

The data can be shared up on request.

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
