# Peer review of "Improving Drug Delivery on Candida Albicans Using Geraniol Nanoemulsion"

_pharmaceutics, 2023, doi:10.3390/pharmaceutics15102475_

Round 1

Reviewer 1 Report

The manuscript presents a very well designed study with appropriate techniques and significant results. My only comments can be found below:

(1) Please add STDEV values also for the standard deviation. 

(2)Figure 2, Figure 4, Figure 5, Figure 7 (keep the red helper line) remove the background lines for all the figures. 

(3) Figure 3 please have the same scale bar and in Figure 6 please add scale bars that are visible for the readers. 

Author Response

Authors thank reviewer 1 for the time dedicated to consideration turning up fully pertinent concerns. Comments are addressed on the following table.

Reviewer 2 Report

Authors proposed a paper entitled “Improving drug delivery on Candida albicans using Geraniol-Nanoemulsion” for the publication in Pharmaceutics.

The paper has a good scientific soundness, and deserves to be published after the following revisions:

Line 19. “C. albicans” should be written in italics.

Line 20. Remove double line before “Nanoemulsions”.

Line 20. “ 10% oil phase” is this on mass basis? Define this also for the surfactant and the aqueous phase.

Line 23. "Zeta”, remove capital letter.

Line 28. Please use dots instead of commas for decimals.

Line 73. “they are … low-cost production,” please rephrase this sentence.

Line 75. Better “appears to be” instead of “NE system appears as an effective”.

Line 90. “with modifications” are these modifications described in the following lines?

Line 112. “2% uranyl acetate solution” is this volume over volume?

Line 125. Can you provide CAS number for reactives?

Line 192. “the analyzed samples” this is still the explanation of the stub preparation; therefore, the samples are not “analyzed”, but are to be analyzed.

Line 233. Did the visual observation confirm that there were no significant variations before the 24 h? According to this, is there any possibility to add the photo before 24 h and after 48 h of observation? These 2 images could be added to Figure 1 and determining a more complete figure.

Line 236. Please remove double space before “Changes…”

Lines 244 and 245 are more explicative about the methods and should be moved to materials and methods section. Please, present directly the results and their discussion.

Line 248. “150.8 ± 1.7 nm, and 232.3 ± 2.7” according to the PDI indicated here, particle size distribution is characterized by a too low standard deviation of about 1-2 nm. Perhaps this is referring to an error and not to a standard deviation of a particle size distribution curve.

Line 262. Remove double space here.

Figure 1a. The ratio among axis here could be misleading.

Line 274. I would define it as “shape” and not “format”.

Table 2. Maybe MFC/MIC could be eliminate from the table and defined only in the caption.

I suggest separating Figure 5 in two different figures, since now the histograms are not clearly visible.

Line 311. “normal” could be substituted with “classic” or to the “reported in literature”. Maybe add some literature references.

Figure 7. Is there the possibility to add errors to these bars?

Line 342. “allows better control of droplet size” please clarify this better control, using quantitative expressions.

Discussion is too long and risks to be only a summary of the already presented results. I suggest moving part of the discussion in the conclusions section, adding also information on future perspectives.

Quality of English needs to be improved in some points, as indicated in my comments.

Author Response

Authors thank reviewer 2 for the time dedicated to consideration turning up fully pertinent concerns. Comments are addressed on the following table.

Reviewer 3 Report

General comments

The submitted manuscript is focused on geraniol based nanoemulsions (NE) to improve its delivery against C. albicans. Thus, its antibiofilm effect and cytotoxicity were investigated.

The topic is interesting, but more recent literature papers have to be introduced, the paper originality has to be highlighted, Materials and Methods section has to be improved, as well as Results one. The Conclusions have to be expanded, in order to evidence the main results.

Some specific remarks and suggestions are reported below point by point.

Abstract

-        The Abstract section is too long and has to be summarised.

1. Introduction

- Corroborate the following statement “Nanocarriers allow a better distribution of drugs to its site of action, a factor that provides a series of benefits to the patient, such as increased therapeutic efficacy, reduced 69 doses, fewer administrations and decreased side effects [14].” with more recent papers, including “Strategies To Improve Ellagic Acid Bioavailability: From Natural Or Semisynthetic Derivatives To Nanotechnological Approaches Based On Innovative Carriers,  Nanotechnology 31[38] (2020): 382001.

- Please, support the following statement “Nanoemulsion (NE) can be formed by dispersing oil in water or water in oil and allow the incorporation of different active substances.” with suitable references, including “Spray dried nanoemulsions loaded with curcumin, resveratrol, and borage seed oil: the role of two different modified starches as encapsulating materials, International Journal of Biological Macromolecules 186 (2021): 820-828” and “Influence of diverse natural biopolymers on the physicochemical characteristics of borage seed oil-peppermint oil loaded W/O/W nanoemulsions entrapped with lycopene. Nanotechnology 32(2021): 505302” and Assessment of Growth Inhibition of Eugenol-Loaded Nano-Emulsions against Beneficial Bifidobacterium sp. along with Resistant Escherichia coli Using Flow Cytometry. Fermentation, 9(2) (2023): 140.

- The aim of the present pare has been reported at the end of the Introduction section, whereas the originality and added value to the scientific community have to be highlighted.

- A list of the used characterisation techniques has to be added at the end of the Introduction section.

2. Materials and Methods

2.1 Nanoemulsion (NE) preparation

More details about all the used reactants and solvents, such as the purity, have to be added.

2.2. Nanoemulsion Characterization

2.2.1. Average hydrodynamic diameter, Polydispersity Index, Zeta Potential

- More details about all the used characterisations have to be added.

3. Results

- In both Figures 5 and 6 the scale bar is not so visible. Please improve it.

4. Discussion

- The concluding considerations, “Our results confirmed the hypothesis that the incorporation of natural compounds into nanostructured lipid systems for drug delivery potentiates their action compared to their free form. Nanotechnology applied to healthcare is constantly growing and arousing increasing interest, based on results of this study, the authors suggest further research on effect of GE and GE-NE on a wide range of pathogenic fungi and bacteria, in addition to in vivo toxicological studies thus confirming efficacy and safety of compound for further clinical application. Future studies with the geraniol release profile will be carried out and may complement this study.”, need to be supported with appropriate references.

5. Conclusions

- The Conclusions section has to be improved and expanded, highlighting the main achievements.

Some moderate revisions are required.

Author Response

Authors thank reviewer 3 for the time dedicated to consideration turning up fully pertinent concerns. Comments are addressed on the following table.

Round 2

Reviewer 2 Report

Authors kindly asked point by point to all my issue raised during the first round revisions.

Authors also provided a new version of the paper, that now deserves to be published in present form.

Reviewer 3 Report

General comments

The paper can be accepted in the current version.

The English language quality is good